# Formative evaluation of the acceptance of HIV prevention Artificial Intelligence chatbots by Black gay, bisexual, and other men who have sex with men in the Southern United States: Focus group study

Jackson Jr Nforbewing Ndenkeh[1], Gloria A. Aidoo-Frimpong[2,3], LaRon E. Nelson[1,2], Mary L. Peng[4], Vimala Balakrishnan[5], Victoria Barnhart[6], Bernard Davis[7], James Donté Prayer[8], Alvan Quamina[9], Zhao Ni[1,2]*

1 School of Nursing, Yale University, New Haven, Connecticut, United States of America, 2 Center for Interdisciplinary Research on AIDS (CIRA), School of Public Health, Yale University, New Haven, Connecticut, United States of America, 3 Department of Epidemiology and Environmental Health, School of Public Health and Professions, University at Buffalo, State University of New York, New York, United States of America, 4 Department of Global Health and Social Medicine, Harvard Medical School, Harvard University, Boston, Massachusetts, United States of America, 5 Faculty of Computer Science and Information Technology, University of Malaya, Malaysia, 6 Community Care Resources of FL (CCRSFL), Pembroke Pines, Florida, United States of America, 7 RAO Community Health, Charlotte, North Carolina, United States of America, 8 The Center for Black Health & Equity, Durham, North Carolina, United States of America, 9 National AIDS Education & Services for Minorities (NAESM), Inc, Atlanta, Georgia, United States of America

* zhao.ni@yale.edu

## Abstract

Gay, bisexual, and other men who have sex with men (MSM) account for 60% of new HIV infections among Black Americans in the Southern United States (U.S.). Despite recommendations for frequent HIV testing and daily pre-exposure prophylaxis (PrEP) uptake, there remains a gap in PrEP uptake among these Black MSM in the Southern U.S. Artificial Intelligence (AI) chatbots have the potential to boost users' health awareness and medication adherence. This study aims to evaluate Black MSM' perspectives on the challenges to the uptake of PrEP and identify Black MSM-preferred chatbot functionalities and platforms for embedding AI chatbots. Five focus group discussions were conducted (February - March 2024) among 21 Black MSM in the Southern U.S. Interview transcripts were thematically analyzed according to challenges to PrEP uptake and the four domains of the Unified Theory of Acceptance and Use of Technology (UTAUT): performance expectancy, effort expectancy, facilitating conditions, and social influence. Black MSM identified lack of awareness or insufficient information, stigmatizations of sexuality, HIV, and PrEP, as well as concerns with side effects, and low self-perceived HIV vulnerability as the major challenges they faced in PrEP uptake. Moreover, chatbots were perceived as an acceptable option for delivering PrEP education (performance expectancy), especially

**Data availability statement:** The quantitative data analyzed, and the exhaustive list of retrieved text reported in this manuscript have been provided as supporting information.

**Funding:** This work was supported by Gilead Sciences, Inc. (IN-US-4126771 to ZN, LN, and VB). JJNN is a Postdoctoral Associate supported by this grant. The funders had no role in study design, data collection and analysis, decision to publish, or preparation of the manuscript.

**Competing interests:** The authors have declared that no competing interests exist.

with accessible, user-friendly interfaces (effort expectancy). Other desired features included simplifying access to PrEP information, incorporating culturally sensitive algorithms, upholding anonymity (social influence), and linking users to healthcare providers and resources (facilitating condition). The study highlights the multifaceted considerations for the adoption of AI chatbots as an HIV-prevention intervention among Black MSM in the Southern U.S.

## Author summary

Due to the higher HIV infection rates among Black gay, bisexual, and other men who have sex with men (MSM), there is an urgent need for interventions to support their decision to access and use pre-exposure prophylaxis (PrEP). The use of an Artificial Intelligence (AI) chatbot can play a role in such an endeavor but their opinion on it matters. We used focus group discussions to get the perspectives of Black MSM on the challenges of using PrEP and what aspects of an AI chatbot would be deemed useful. We found that there was insufficient information on PrEP to guide decision on its uptake, coupled with issues around stigmatization, concerns about side effects and low perceived risk. We also found that chatbots can be useful in PrEP uptake but its acceptance would be linked to its ability to deliver adequate PrEP/HIV education, have easy to use interfaces as well as converse in a culturally sensitive manner, and improve clients' access to healthcare services. Our study provides insights on key aspects to consider for the use of AI chatbots in HIV prevention among Black MSM in Southern region of United States.

## Background

HIV remains a significant global public health challenge, affecting approximately 40 million people worldwide [1]. In the United States (U.S.), HIV continues to be a critical health issue, with over 1.2 million people living with HIV [2]. The Southern region of the country bears a disproportionate share of the burden, accounting for 49% of new HIV diagnoses in 2022, [3] despite representing only about 38% of the U.S. population [4,5]. Black Americans in the South are particularly impacted by HIV, representing half of all new HIV diagnoses in the region in 2021, despite comprising only 19% of the Southern population [4]. Among them, Black gay, bisexual, and other men who have sex with men (MSM) are especially vulnerable. In 2022, Black MSM accounted for about 35% of new HIV diagnoses among all MSM in the U.S., with a significant concentration in Southern states [3] Black MSM have a one in three lifetime risk for HIV infection, [6] facing multiple barriers to HIV prevention and care, including stigma, discrimination, and socioeconomic disparities [2,7]. These factors contribute to their increased vulnerability and adverse outcomes in the HIV prevention and care continuum.

Despite an overall stabilization of HIV diagnoses among Black MSM, those in the South continue to experience unfavorable outcomes [2–4,7]. The Health and Human Services (HHS) initiative, "Ending the HIV Epidemic: A Plan for America," highlights the critical role of biomedical advances in HIV prevention, including pre-exposure prophylaxis (PrEP), in reducing new infections [8]. Equitable access to PrEP is crucial for reducing HIV infections among Black MSM. PrEP, a medication taken by HIV-negative individuals to prevent infection, has proven highly effective [9]. However, its uptake among Black MSM remains low due to factors such as lack of awareness, mistrust in the healthcare system, stigma, and limited access to culturally competent healthcare providers [10,11].

The integration of digital technology in healthcare has transformed the prevention and management of chronic diseases, including HIV. Digital health interventions such as mobile health applications (mHealth apps), telemedicine, and online health platforms have significantly expanded access to healthcare services and information, especially for underserved populations [12–14]. These technologies help overcome barriers to HIV prevention and care by delivering timely, accurate, and personalized health information to those at risk [14].

Artificial Intelligence (AI) is emerging as a transformative tool in the digital health landscape, offering innovative solutions for HIV prevention and care [15,16]. In the context of HIV prevention, AI chatbots and virtual health assistants can play a crucial role in enhancing awareness, education, and communication about PrEP. These AI tools can engage users in interactive conversations, answer questions, provide reminders for medication adherence, and offer support in a confidential and non-judgmental manner [17]. Additionally, AI chatbots have the potential to provide culturally competent and linguistically appropriate information about PrEP, addressing common misconceptions and reducing the stigma associated with its use [18]. These chatbots can also facilitate linkage to care by providing information on local healthcare providers, clinics, and support services [19].

AI chatbots have emerged as promising tools in healthcare, particularly for managing chronic conditions and improving patient engagement [20]. In the realm of HIV prevention, AI chatbots can play a pivotal role by delivering personalized, stigma-free interventions, promoting HIV testing, and supporting PrEP adherence, with sophisticated data analysis and predictive analytics. For instance, a study conducted in Malaysia demonstrated the feasibility and acceptability of using an AI chatbot to promote HIV testing and PrEP uptake among MSM [21]. The chatbot provided comprehensive information on HIV testing and PrEP, and users found it easy to use and helpful in avoiding stigma-inducing interactions, thus potentially increasing the frequency of HIV testing and PrEP uptake [22].

To design AI chatbots that are accepted by Black MSM and deemed useful for HIV prevention, it is necessary to learn from their perceptions and expectations of AI chatbots. We need to align the AI chatbots with their HIV prevention context and consider features that will ensure their interest and engagement. In this light, we conducted this study to learn participants' perspectives on the challenges to PrEP uptake and to identify chatbot functionalities and platforms preferred by Black MSM.

## Methods

### Study design and inclusion criteria

This was a qualitative study conducted through five focus group discussions (FGDs) which took place between February and March 2024 among 21 Black MSM (3–5 per group) who reside in the Southern U.S. Inclusion criteria for the participants were as follows: (1) cisgender male, (2) age ≥ 18 years, (3) having Internet access, (4) speaking English, and (5) self-reporting condomless sex with another man in the past 6 months.

### Participant recruitment

Participants of this study were recruited through convenience sampling using a web-based eligibility screener that included questions about the aforementioned inclusion criteria. The screener was posted by a research assistant (RA) on

social networking apps commonly used by MSM for dating, including Grindr, Jack'd, Hornet, and WhatsApp groups. Once a person filled the screener and was deemed eligible, the RA reached out to them providing details about the study and answering any questions that came up for better understanding. The RA then provided the opportunity for participation documented through electronic consenting of participant. A total of 21 individuals initiated the screening process, and all 21 were deemed eligible to participate and consented to join the focus group discussions.

## Data collection procedures

Upon providing informed consent, participants were assigned to groups based on their availability for FGDs. The FGDs were conducted using Zoom (Zoom Video Communications), where the video setting was turned off and participants were instructed to use pseudonyms. The FGDs were conducted by an experienced research assistant (RA) under the supervision of the corresponding author, using an interview guide. This interview guide was tailored from another study conducted in Malaysia[21] and comprised of two main discussion sections. The first section aimed to gather their perspectives on the current barriers they face with respect to PrEP uptake and constituted questions like "what gets in the way of accessing or taking PrEP?" and "what can be done to improve access and/or uptake of PrEP?" The second section, building on the first, aimed to gather their perspectives on the usefulness of AI chatbots in HIV prevention, identifying features that could enhance user engagement, and exploring which digital platforms would be ideal for integrating these chatbots. It constituted questions like "what do you think about the use an AI chatbot to promote PrEP awareness and uptake?", "what chatbot features do you think will encourage its acceptance and use?" and "On what platforms would you like to see the above chatbots integrated?" Some of these questions had incorporated prompts to initiate discussion among study participants. Also, as part of the screening process, each participant was requested to complete a brief survey constituting Yes/No or nominal categorical questions on their demographic characteristics, self-reported substance use, depressive symptoms (PHQ-2) and HIV testing. To foster honesty in participants' responses, we insisted on the respect of the opinion of one another while underlining that there were no wrong or right answers, just diverse opinions all of which were welcomed. After each FGD session the RA and corresponding author had a debriefing where they discussed on the points raised during the session and agreed on whether there was need to continue with the next FGD session (or that saturation was achieved) based on repetitiveness of themes and diversity of opinions.

## Conceptual framework for analysis

To assess Black MSM's perspectives on the challenges to the uptake of PrEP, we conducted a thematic analysis of FGD transcripts. This analysis not only identified challenges but also revealed possible ways to increase PrEP uptake from the participants' viewpoints. Additionally, we used the Unified Theory of Acceptance and Use of Technology (UTAUT) to guide the analysis of MSM's acceptance of AI chatbots for HIV prevention. The UTAUT is a valid, reliable, and widely used framework in the field of digital health. It has been applied among Black MSM in the Southern U.S [23]. Our team has also used it to measure MSM's acceptance of AI chatbots in Malaysia [21,22]. Applying a uniform framework allows us to compare the insights of MSM from Malaysia and the Southern U.S. for future research by ensuring that data are collected and analyzed in a consistent manner across both contexts, enhancing the validity of cross-regional comparisons and minimizing potential biases introduced by varying methodologies. Pulled from the Technology Acceptance Model (TAM), which had only two constructs; perceived usefulness and perceived ease of use, the UTAUT further expands TAM to have four constructs namely: performance expectancy (perceived usefulness), effort expectancy (perceived ease of use), social influence (the influence of family, peers and friends on users' acceptance and/or intention to use an app), and facilitating conditions for adoption and use of technology [21,23]. Venkatesh et al. proposed thus the UTAUT as a unified model for user acceptance of information technology, and demonstrated that the above four constructs played significant roles as direct determinants of client intention to use and their subsequent use of information technology [23]. The framework

focuses on the perspective of the potential users on what and how the technology will be useful to them rather than technology developers' and researchers' assumptions [21].

### Analyses

The data collected from study participants were analyzed using R version 4 for quantitative analysis and Provalis Research QDA Miner Lite version 3 for qualitative analysis. Descriptive statistics were used to summarize participants' sociodemographic characteristics, presented as both absolute and relative frequencies. The FGD transcripts were analyzed using thematic coding, where key emerging themes were pooled at two levels. At the first level, key themes were pooled with respect to MSM's opinions on the challenges of PrEP uptake and their perspectives on possible ways to increase PrEP uptake. At the second level, key themes were pooled with respect to participants' perspectives on the benefits, concerns, and desirable features of the AI chatbots. The first round of thematic coding was conducted by JJNN and ZN, after which they discussed their respective findings and resolved all discrepancies in the emerging themes. These were then presented to other co-authors (LEN and GAF) at the second level for another round of review. When the aforementioned individuals reached a consensus on the key emerging themes, JJNN then proceeded to consolidate and organize the quotations from the second level of thematic coding according to the four UTAUT constructs.

### Ethical consideration

This study was approved by the Yale University Institutional Review Board (ID: 2000035242). All procedures involving human participants were in accordance with the ethical standards of the United States and Yale University as well as with the Helsinki declaration. The FGDs were conducted only for persons who had voluntarily consented to participate in the study, after duly receiving information on study-related risk and benefits. Each participant was compensated $25 for completing the FGD.

## Results

### Participant characteristics

Participants' ages ranged from 28 to 60 years, with nine participants (42.9%) between the ages of 36 and 45. Ten participants (47.6%) had a history of substance use within the six months preceding the study, and six participants (28.5%) consumed six or more alcoholic drinks at least once per week. In the two weeks prior to the study, four participants (19%) experienced feelings of sadness, depression, or hopelessness for several days, or reported diminished pleasure in activities for more than half of the days. Lastly, sixteen participants (76.2%) had tested for HIV before, among which three (14.3%) had received a positive HIV diagnosis (Table 1).

### Participants' perspectives on challenges to PrEP uptake

**Lack of awareness or insufficient information on PrEP.** In our FGDs, participants unanimously agreed that there was a general lack of awareness of PrEP and its benefits within the Black MSM communities in the Southern U.S. Participants noted that many individuals had not even heard of PrEP, let alone understood its benefits. One participant remarked, "*not enough information is going around, and people don't promote it (PrEP) as much*", suggesting that insufficient promotion contributed to low awareness. This sentiment was reinforced by others who pointed out that even when people encountered advertisements for PrEP, they often lacked the educational content necessary to inform decision-making. As one participant explained, "…*they say that PrEP is available, but people don't really understand, ok yeah, it's available, but what does that even mean?*" This lack of clarity extends to understanding when and how to properly use PrEP. Furthermore, participants expressed that healthcare professionals have not been sufficiently proactive in educating and informing their patients about PrEP. One participant shared, "*sometimes when you go to certain doctors*

**Table 1. Participants' characteristics (N = 21).**

| Variable | n (%) | Variable | n (%) |
|---|---|---|---|
| *Age group* | 21 | *Little pleasure in doing things* | 21 |
| 28-35 years | 7 (33.3) | Not at all | 14 (66.7) |
| 36-45 years | 9 (42.9) | Several days | 3 (14.3) |
| 46-60 years | 5 (23.8) | More than half of the days | 4 (19) |
| *Substance use in the past 6 months\** | 21 | *Feeling down, depressed, or hopeless* | 21 |
| No | 11 (52.4) | Not at all | 17 (81) |
| Yes | 10 (47.6) | Several days | 4 (19) |
| *Frequency of drinking alcohol* | 21 | *Tested for HIV* | 21 |
| Never | 2 (9.5) | No | 5 (23.8) |
| Monthly or less | 8 (38.1) | Yes | 16 (76.2) |
| Two to four times a month | 5 (23.8) | *HIV test results* | 21 |
| Two to three times a week | 4 (19) | Negative | 13 (61.9) |
| Four or more times a week | 2 (9.5) | Positive | 3 (14.3) |
| *Frequency of six or more alcohol drinks* | 21 | Unknown | 5 (23.8) |
| Never | 14 (66.7) | *Number of alcohol drinks on a typical day* | 19 |
| Monthly | 1 (4.8) | One or two drinks | 14 (73.7) |
| Less than monthly | 3 (14.3) | Three or four drinks | 3 (15.8) |
| Weekly | 2 (9.5) | Five or six drinks | 1 (5.3) |
| Daily or almost daily | 1 (4.8) | Ten or more drinks | 1 (5.3) |

\* Substances included heroin, morphine, benzodiazepines, marijuana, crystal methamphetamine, ecstasy, ketamine, poppers, GHB, and steroids.

*or hospitals, they don't even talk about it (PrEP).*" This highlights a critical gap in patient education within healthcare settings.

Participants emphasized the need for increased sensitization, education, and communication about PrEP and sexual health, underscoring the importance of providing sufficient educational content to aid decision-making. They suggested that these efforts should involve healthcare professionals who regularly interact with community members and extend to community events where Black communities gather, such as sporting events, clubs, and bars. One participant summarized this approach, "*… you have a billboard, but will people actually take the time to really do that investigation based off of a billboard? No, you need to go to the people, go to the community where they are, and the thing about it is, there are people out there that will love for us to go out there in the community and to talk about the vitality of PrEP*". This indicates a preference for direct community engagement over passive advertising. Another participant recalled a past event where the LGBTQ+ community was represented "*…I worked at the stadium that day, and it was a massive amount of people! And a lot of the gay community came out in huge drove, and I didn't see like the vans or any of the people promoting that type of stuff, and I think that would have been a nice place to help make it more normal.*" This suggests that integrating PrEP education into large community gatherings could normalize its use and increase uptake. The FGDs reveal that the primary barriers to PrEP uptake among Black MSM in the Southern U.S. stem from a lack of awareness and insufficient information. The repeated emphasis on direct engagement suggests that passive advertising methods are less effective in reaching this demographic.

**Stigmatization of sexuality, HIV and PrEP.** Participants believed that stigmatization towards PrEP, sexuality, and HIV has significantly contributed to their reluctance to adopt PrEP as a prevention method. One participant reflected on the historical context, noting that the advent of the HIV epidemic linked HIV diagnoses to promiscuity and inevitable death, a narrative persisting particularly in the Black community where religious and cultural values have been passed down

PLOS Digital Health

through generations. A participant explained, *"…them thinking that me getting on PrEP is me acknowledging I have sex a lot… and I think it goes back to the stigma; nobody wants to be known as promiscuous."* Another participant echoed this by narrating an experience with friends, *"I got a few friends…, they were in my room, I left the (PrEP) bottle on my nightstand, and they saw it, and I saw the looks."* Furthermore, participants noted that PrEP advertisements often target the gay community, reinforcing the stigma. As one participant stated, *"there's a stigma attached to it (PrEP); that it's only for gay people, and I don't want people to know that I'm gay."* For most participants, going for PrEP has been associated with judgment, ridicule, and embarrassment.

To address these issues, participants stressed the importance of dismantling the stigma associated with sexuality, HIV, and PrEP. They suggested creating safe spaces where individuals can comfortably discuss these topics and combat stereotypes. Regarding PrEP-related stigma, participants agreed that messaging should be more inclusive and not solely geared toward specific populations. As one participant mentioned, *"…it shouldn't just be for the LGBTQI, all of that it should be for the straight people as well"*, while another said, *"there should be this advertisement to make it more normal… anybody can catch HIV so it should be everywhere"*. Overall, the results highlight that stigma and misinformation are significant barriers to PrEP uptake among Black MSM in the Southern U.S. Historical context and cultural values continue to influence perceptions, making it difficult to normalize PrEP use. There is a clear need for inclusive messaging address these stigmas directly.

### Concerns on side effects and general reticence towards drugs and vaccines

There are concerns about the side effects of the PrEP regimens, which are often intensified for those who have already faced a lengthy decision-making process, making the fear of side effects a significant obstacle in broader PrEP adoption. A participant said *"…and the side effects was a lot for me, so I think once people experience that, it's kind of like, I don't want deal with that. … and just to be safe, go through all of these side effects, …they don't see it as a benefit."* This highlights a key barrier to PrEP uptake, as potential users weigh the benefits against the risks of side effects.

Moreover, there is a general reticence among Black men about hospitals, medications, and vaccines, stemming from a distrust of a healthcare system that has historically failed to promote equity. A participant said *"…how Black people, … was being pretty much experimental bodies for the government, so for me, that's one of my hesitations, …one of those (challenges) would be just like trusting that these people are actually making medicine that is actually good for us and that it's not going to harm us in the long run."* This historical context of medical mistrust significantly impacts attitudes towards PrEP.

To address these concerns, participants highlighted the importance of education and providing detailed guidance for informed decision-making about PrEP. As one participant suggested, *"more health fair set is dedicated to Black men, … it will bring more awareness."* Another participant backed this up by saying *"Black men don't go to the hospital, unless when they are in pain or in an ambulance..."*. Overall, the discussions reveal that concerns about side effects and a deeply rooted mistrust of the healthcare system are substantial barriers to PrEP uptake among Black MSM.

### Perceived vulnerability or contentment with condom use

Participants noted that self-perceived low HIV risk and satisfaction with condom use were additional factors influencing their decision to take PrEP, albeit to a lesser extent. One participant stated that *"young people also think they're invincible, like nothing is going to happen to them"*, and another participant, echoing the sentiment, stated, *"I think Black, gay men just cannot recognize that they are promiscuous, …they can't acknowledge that they have a colorful sexual lifestyle"*. The above *"colorful sexual lifestyle"* can be a way of dealing with mental health problems as mentioned by a participant, *"Black people are not the first ones to go get mental health checkups and stuff like that, …so they deal with things, how they want to deal with it. So that's why some people are promiscuous, and they just do whatever."* Furthermore, some participants indicated a preference for condom use as their primary method of protection. One participant explained, *"I use*

*a condom that'll be like the go to… I guess they were feeling like PrEP is a license to not use condoms.*" This suggests that some individuals view condoms as sufficient protection and see PrEP as unnecessary or as promoting risky behavior. These insights underscore the complexity of attitudes towards PrEP uptake among Black MSM. The perception of invincibility, especially among younger individuals, significantly reduces the perceived need for additional protective measures like PrEP. Additionally, the reluctance to openly acknowledge sexual behavior, along with the use of sexual activity as a coping mechanism for mental health challenges, further complicates the adoption of PrEP. A reliance on condoms, viewed by some as entirely adequate, reflects the belief that PrEP might encourage riskier sexual behavior.

### Participants' perspectives on preferred features

The preferred features of AI chatbots for HIV prevention from our study participants are presented in line with the UTAUT used to guide the analysis of MSM's acceptance of AI chatbots for HIV prevention (Fig 1).

### Performance expectancy

Participants had mixed perceptions regarding the performance expectancy of chatbots. The consensus was that AI chatbots could be an acceptable option for disseminating information on PrEP, safe sex, and HIV, especially to younger generations who are more engaged with mobile technology. One participant noted, "*…it really couldn't hurt. I mean, … it's another form of communication …you would reach a younger generation, and you know even the older generation…*". However, some participants were concerned about the chatbots' ability to provide emotional support. As one participant mentioned, "*I feel like it depends on how you look at being helped by somebody who's not human. … even when I'm doing something that's just say with my phone or something. Perhaps if I need like some help, … I'd rather speak to you on the phone so I can hear your emotions. I can figure out how you're responding to my situation.*"

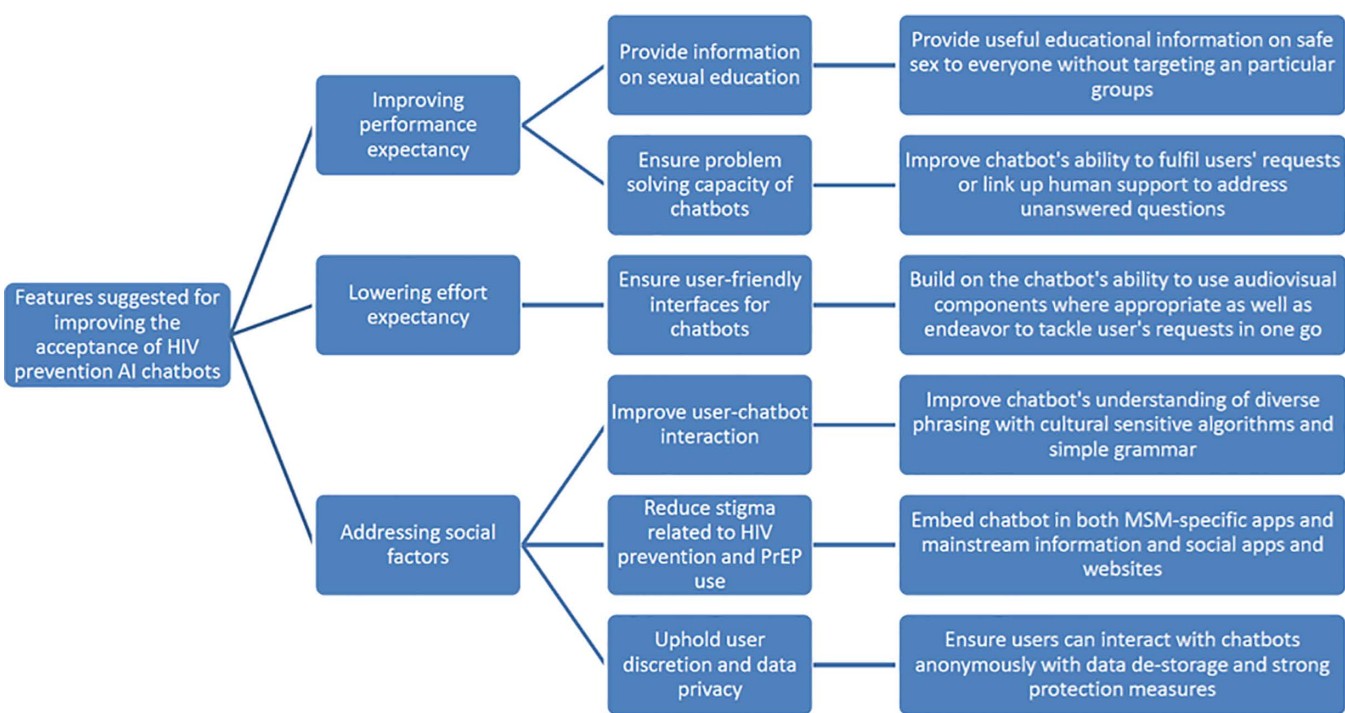

**Fig 1. Features suggested to improve acceptance of HIV prevention AI chatbots (N = 21).**

Participants highlighted two main features that would improve chatbot performance, including the ability to provide useful health information and effective problem-solving capabilities. Participants indicated that AI chatbots should offer educational information on safe sex to both the gay community and the general population. One participant said "*…I just wouldn't put them on the sexual situation, I mean kind of on a general basis of information of this all around health, … especially with anyone who's having sex, I mean like anyone.*" The chatbot's problem-solving abilities were emphasized in two scenarios: first, its capacity to fulfill users' requests for resources (e.g., HIV test kits, PrEP), and second, its ability to connect users with someone who could address requests or answer questions that the chatbot could not resolve. A participant said "*…if it was able to fulfill orders or requests for (HIV) test kits, …whatever the situation calls for, that will be a great asset to the chatbot other than just, …vaguely answering questions.*" Another participant said "*…and then, if you want something further, and to speak to somebody, it (chatbot) can capture that and send that information over to whoever, so that the person can be contacted in that space.*" Another participant suggested incorporating hyperlinks that could easily direct users to local resources for accessing PrEP, tailored to their financial situation, for instance "*…where they can get a coupon from to get the medication (PrEP) if they don't have any type of insurance.*" The varied perceptions of chatbots' performance expectancy reflect a broader conversation about the role of technology in healthcare communication. While chatbots are seen as valuable tools for reaching younger, tech-savvy generations, their ability to provide the necessary emotional support remains a concern. Participants' focus on the importance of providing useful information and effective problem-solving underscores the need for chatbots to be thoughtfully designed to address these requirements comprehensively.

### Effort expectancy

Participants had mixed perceptions of effort expectancy regarding AI chatbots. Their positive perceptions were primarily linked to the convenience of using chatbots, especially for discussing sensitive topics privately. One participant noted "*I think it could potentially be helpful for people who are trying to get information anonymously. People who may not be educated but want to ask questions or just kind of get like streamlined information, they would have an ability to do so.*" This highlights the potential of chatbots to provide a discreet and accessible platform for individuals seeking information on sensitive subjects.

However, participants also pointed out several technical problems associated with existing chatbots. These issues, often stemming from the underlying algorithms, include the chatbots' inability to answer certain questions or providing repetitive responses. As one participant explained, "*…some questions they don't answer, and some questions they repeat the same answer, because they're only able to respond certain questions.*" This can be particularly frustrating for users who are in an urgent need of those responses, as evidenced by a participant who stated, "*I get frustrated with chatbots as well as automated services …just like, okay, I need to talk to somebody, because this is irritating.*"

To improve chatbots' effort expectancy, participants emphasized on the need for user-friendly interfaces. One notable element that emerged was the use of audiovisual features for engaging users who prefer them over text-based interactions. A participant said "*…what I would suggest is having a slight video of what PrEP does in the body, …so that they can have an idea of what it does in the body. It's one thing when somebody can say, talk about it, by word of mouth but when you actually see it in the video, it kind of makes more sense.*" Another element was the importance of designing chatbots to handle users' requests efficiently in a single interaction. In that light, a participant said, "*If the matter or issue could be resolved in a one call situation as opposed to having to hang up or get out of that chat to keep asking the same question over and over.*" The participants' feedback highlights both the potential benefits and challenges of using chatbots in healthcare communication. On the positive side, chatbots offer a convenient and discreet way for individuals to seek and disclose information about sensitive topics, such as PrEP, HIV, and sexual activities. This anonymity can encourage more people to seek help without fear of judgment. However, the current limitations in chatbot technology, such as repetitive and unhelpful responses, can hinder their effectiveness and frustrate users.

**Facilitating conditions**

To facilitate the adoption of AI chatbots for HIV prevention, participants emphasized the importance of maintaining a human component through efficient linkage to healthcare professionals based on user needs. One participant noted "*…initially, I think it is a good idea, because, you know, I think it's there to kind of keep tabs on people, but once we do grab their attention, …you want to kind of direct them to a real person where they do feel comfortable to kind of like even it out…*". This underscores the need for chatbots to transition users to human support when necessary, ensuring a seamless experience.

Participants also emphasized the importance of having the system monitored by a human operator who can intervene if the interaction does not progress as expected. One participant compared this to customer service AI, which can be irritating if not properly managed. They said, "...*sometimes can be irritating."* Another participant highlighted that reducing the user's awareness of interacting with AI might increase adoption: *"The less the user feels like they are talking to AI, the more they'll be inclined to adopt it."* This suggests that chatbots need to be highly adaptable and responsive to user inquiries, providing personalized and human-like interactions. Participants' feedback highlights many crucial factors for the successful implementation of AI chatbots in HIV prevention. The need for a human element is paramount, with chatbots acting as an initial point of contact that can seamlessly transition users to healthcare professionals for more personalized support. This hybrid approach can help build trust and comfort, which are essential for user engagement. Additionally, human oversight is recommended for quality control and timely intervention when interactions fall short of expectations. Lastly, enhancing chatbot adaptability and responsiveness to specific questions can help reduce the perception of interacting with AI, encouraging greater user adoption.

**Social factors**

Participants discussed three key social factors shaping the adoption of chatbots for HIV prevention, including linguistic adaptation, anonymity and privacy, and tailored embedded platforms. First, some participants raised concerns about chatbots' technical ability to understand, relate to, and engage effectively with users, specifically concerns about whether a chatbot can understand and respond to questions from diverse populations with different dialectical and linguistic features across different geographical locations. One participant suggested conducting diverse community-based surveys to gather questions that people commonly ask about PrEP. These questions could then be integrated into the chatbot's algorithm, ensuring that it not only recognizes the questions but also understands the various ways they may be phrased. Another participant emphasized the importance of simplifying interactions by using straightforward language, avoiding complex medical jargon.

Second, in terms of discretion and privacy, participants suggested it would be beneficial for the chatbots to allow anonymous interactions, ensuring that no data is stored or shared. This would prevent others from accidentally accessing the conversation and safeguard users' privacy from the app developers. As mentioned by one participant, "*…as long as it stays anonymous and discreet, everyone loves anonymous and discreet.*" Likewise, another participant stated, "*…keep that in a little private area where you don't want nobody to know that you're on the app, so you get some discretion, but you're still getting your information.*"

Lastly, participants emphasized the importance of integrating chatbots into both MSM-specific apps and apps for the general population, as a strategy to reduce PrEP-related stigma. When asked about preferred platforms, participants mentioned not only MSM apps like Jack'd, Grindr, and Scruff, but also mainstream apps such as Tinder, Instagram, Facebook, Snapchat, and Twitter. One participant explained, "*Jack'd, Grindr, Tinder, …the biggest places where people meet up and, you know, have sex, or hook up, or whatever the case may be*". Another participant added that in "*…trying to get everybody…*", the AI chatbots should also be embedded on websites (for example, Sniffies) accessible on computers rather than focusing only on mobile apps for "*some people like to be on the Internet, on a computer*".

The discussions highlight the significance of social factors in the adoption and effectiveness of AI chatbots for HIV prevention. Ensuring that chatbots can understand diverse linguistic expressions and provide relatable and easily

understandable communication is crucial for user engagement. Privacy and discretion are paramount, with participants emphasizing the need for anonymous interactions to prevent data breaches and data de-storage to ensure confidentiality. To further reduce PrEP-related stigma, chatbots should be integrated across a wide range of platforms, including both MSM-specific and general population apps, as well as various websites. This broad integration strategy can help normalize PrEP discussions and reach a wider audience, ultimately enhancing the overall effectiveness of HIV prevention efforts.

## Discussion

### Principal findings

To develop an AI chatbot that effectively engages Black MSM and produces the desired preventive effect among this vulnerable population, it is important to understand the challenges they face with PrEP and HIV prevention and how those challenges intersect with their preferred AI chatbot features. Our study participants showed a consensus that an AI chatbot is a viable and acceptable tool for promoting knowledge about PrEP, HIV, and sexuality, with its usability and scalability mainly depend on three factors: providing sufficient information, avoiding stigma, and protecting privacy. To promote PrEP uptake using an AI chatbot, the chatbot needs to be able to adequately interact with users to improve their awareness and willingness to use PrEP by providing sufficient information to guide their decision on the use of PrEP. Considering the extensive stigma associated with HIV, gender/sexual orientation, and PrEP messaging (geared towards MSM), our participants emphasized the critical role that AI chatbots can play in destigmatizing PrEP, HIV, and sexuality, thus improving HIV prevention. A key factor that influences the usability of an AI chatbot is whether it can protect participants' privacy using a trustworthy method and gain users' trust. Embedding the chatbot into a healthcare system that is well-protected by a secure cloud environment is important. In addition to upholding user discretion and privacy, our participants also suggested that PrEP normalization can be achieved by targeting PrEP messaging to wider populations and embedding AI chatbots in popular apps for the general population.

Consistent with other studies, [21,22,24,25] this study's findings have shown that AI chatbots can be a viable and useful tool for HIV prevention, particularly for younger generations who are increasingly inclined towards using mobile technologies for their daily routines [22,24,25]. The potential of leveraging chatbot technology for HIV prevention, especially in raising awareness, education, and communication is immense. Integrating AI and chatbot technologies can enhance the personalization of health messages, ensuring they resonate with users' unique circumstances and needs, as well as enhance engagement, thus increasing the likelihood of adopting desired health behaviors, such as PrEP uptake and adherence [26,27]. For example, AI-powered chatbots and virtual health assistants can provide real-time responses to questions about PrEP, reminders to take medication, and motivational messages to encourage adherence [28]. These AI chatbots can thus be particularly beneficial in addressing the unique needs of Black MSM facing stigma and discrimination, by offering a confidential and supportive environment for managing their health. However, whether such an AI chatbot can be scaled in real-world settings depends on the chatbots' problem-solving ability, ease to use, and user's trust of an automated system [29,30].

According to our study participants, it is essential for developers to ensure that chatbots provide valuable educational information through user-friendly interfaces and are equipped to meet user requests for resources, such as HIV test kits. A similar study in Malaysia underlines the ability of AI chatbots to provide useful health information and to help solve problems of users to be indicative of their level of usefulness [21,22]. Moreover, some studies with AI chatbots in customer services have underlined that the visual design of a chatbot's interface in terms of layout, font and colour, which should be simple, intuitive and appealing, can significantly impact perception as well as encourage user interaction [30,31]. Another key feature of AI chatbots, highlighted by our study participants for user engagement, is their ability to exhibit human-like communication traits, such as natural language use and appropriate expressions of emotions and empathy, with an appropriate balance between overly robotic and overly human-like [21,31]. To achieve this, end users should be actively involved in the chatbot development process. Their feedback, particularly during the testing phase and in subsequent

updates after launch, is crucial to creating a product that is not only functional but also engaging and well-received by users [30].

The use of AI chatbots for education and communication is especially important in contexts where sufficient information about PrEP is not widely available. This is the case with this study, where participants underlined a lack of awareness of PrEP and its benefits within Black communities in the Southern U.S. Even when receiving advertisement about HIV preventive measures, participants mentioned that there was not enough information to guide decision-making on PrEP uptake. This finding is consistent with other research, which highlights the need to increase awareness and availability of PrEP information among populations vulnerable to HIV [32–34].

AI chatbots can play a role in mass education and communication with direct community engagement, as opposed to passive advertising. The participants' insights underscore the need for more targeted and comprehensive educational efforts, both within healthcare settings and at community events. Providing comprehensive information about PrEP not only improves knowledge and awareness of PrEP but also helps reduce PrEP-related stigma, addresses misconceptions about its use, discusses concerns about side effects, and informs the decision to use it, thus improving its uptake [33,35,36]. This aligns with the Information–Motivation–Behavioral (IMB) Skills Model, which posits that when vulnerable populations are well-informed about PrEP, motivated to apply that knowledge, and equipped with the necessary behavioral skills to access it, they are more likely to adopt its use [33]. It should further be noted that to effectively convey information on PrEP among Black MSM, it is important to enhance user experience through embedded features within chatbots' interfaces that lower effort. For instance, quick reply buttons and the use of audiovisual components, among other features, can significantly improve the user experience [37]. In addition, it's important for AI chatbots to identify potential candidates and seamlessly link them to health care personnel for HIV testing, PrEP services (including programs for individuals without health insurance), or other services, notably mental health, substance use, and social services.

This study further highlights that PrEP messaging, when perceived as exclusively targeting the gay community, may lead individuals to feel stigmatized or labeled as promiscuous for using PrEP. Golub *et al*. explain that the promiscuity aspect of PrEP-related stigma is associated with the perception, before the advent of PrEP, that "safe" sexual behavior could only be achieved through limiting sexual activity and/or partners and consistently using condoms [38]. For that reason, persons who use PrEP are falsely considered to be engaging in "risky" sexual behaviors even when they are not. [38] This PrEP-related stigma intersects with existing stigmas surrounding HIV and sexual orientation already faced by the Black MSM population, further hindering their PrEP uptake [39–41]. Our study participants proposed that there is a need to rethink the communication strategy on PrEP specifically, and on HIV/sexuality more generally. One such strategy that can be powered by AI chatbots is the creation of online safe spaces where Black MSM can comfortably discussing HIV, sexuality, and PrEP, as well as dismantling related stereotypes.

Our participants highlighted that to enhance AI chatbots' overall efficacy in HIV prevention, the chatbots must be capable of providing user-centered, reliable, and up-to-date information on PrEP, as well as connecting users to additional resources, so that PrEP can be demystified [27,42]. Another way to leverage AI chatbots would be to widen the target population for PrEP messaging by integrating AI chatbots into both MSM-specific apps and other popular apps geared towards the general population. While we strive for a world where PrEP, HIV, and sexuality are destigmatized, the need for privacy and confidentiality cannot be understated. AI chatbots must provide trustworthy platforms for users to seek knowledge on PrEP without fear of privacy or confidentiality breaches, judgment from the interlocutor, or disclosing their sexual orientation in conservative societies [27,43].

### Limitations

Given the convenience sampling method used to recruit study participants, the findings of this study should be interpreted with several limitations in mind. First, the research assistant resides in Florida, where most participants were recruited, which may limit the geographic diversity of the sample. Second, our sample size is relatively small, thus limiting its

generalizability to all Black MSM in the Southern U.S. However, the challenges to PrEP uptake discovered in this study align well with other study findings, and this study provides valuable insights into how AI chatbots can be leveraged to promote HIV prevention among Black MSM. Moreover, it offers a detailed exploration of key concerns from the perspective of Black MSM regarding the use of AI chatbots for HIV prevention, which is essential for the effective development and implementation of such technologies in this population.

## Conclusion

The study highlights the multifaceted considerations for the adoption of AI chatbots as an HIV-prevention intervention among Black MSM in the Southern U.S. The use of AI chatbots is an acceptable means of communication for behavioral change among Black MSM in the Southern U.S. in relation to HIV prevention in general and PrEP uptake in particular. To promote usability, the development of these AI chatbots should place emphasis on the chatbots' ability to provide sufficient health information, interact adequately with users, and address their concerns while ensuring their privacy. Additionally, gearing AI chatbots' messaging to a wider audience could help demystify PrEP and normalize its use.

## Supporting information

**S1 Text.  Interview Guide.**
(DOCX)

**S1 Data.  Survey database.**
(XLSX)

**S2 Text.  Retrieved texts.**
(XLSX)

## Acknowledgments

We thank Lorenzo Davis for his assistance in recruiting participants and conducting focus group interviews. We also sincerely thank all participants for their courageous and transparent input and insights.

## Author contributions

**Conceptualization:** Jackson Jr Nforbewing Ndenkeh, Gloria A. Aidoo-Frimpong, LaRon E. Nelson, Mary L. Peng, Zhao Ni.

**Data curation:** Jackson Jr Nforbewing Ndenkeh, Gloria A. Aidoo-Frimpong, Zhao Ni.

**Formal analysis:** Jackson Jr Nforbewing Ndenkeh, Gloria A. Aidoo-Frimpong, LaRon E. Nelson, Zhao Ni.

**Funding acquisition:** LaRon E. Nelson, Vimala Balakrishnan, Zhao Ni.

**Investigation:** LaRon E. Nelson, Victoria Barnhart, Bernard Davis, James Donté Prayer, Alvan Quamina, Zhao Ni.

**Methodology:** Zhao Ni.

**Project administration:** Jackson Jr Nforbewing Ndenkeh, LaRon E. Nelson, Zhao Ni.

**Supervision:** Zhao Ni.

**Validation:** Jackson Jr Nforbewing Ndenkeh, Gloria A. Aidoo-Frimpong, LaRon E. Nelson, Zhao Ni.

**Writing – original draft:** Jackson Jr Nforbewing Ndenkeh, Gloria A. Aidoo-Frimpong, LaRon E. Nelson, Mary L. Peng, Zhao Ni.

**Writing – review & editing:** Jackson Jr Nforbewing Ndenkeh, Gloria A. Aidoo-Frimpong, LaRon E. Nelson, Mary L. Peng, Vimala Balakrishnan, Victoria Barnhart, Bernard Davis, James Donté Prayer, Alvan Quamina, Zhao Ni.

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
