## [Decision Letter · Decision Letter 0]

26 Feb 2025

PDIG-D-24-00512Formative Evaluation of the Acceptance of HIV Prevention Artificial Intelligence Chatbots by Black Gay, Bisexual, and Other Men Who Have Sex with Men in the Southern United States: Focus Group StudyPLOS Digital Health Dear Dr. Ndenkeh, Thank you for submitting your manuscript to PLOS Digital Health. After careful consideration, we feel that it has merit but does not fully meet PLOS Digital Health's publication criteria as it currently stands. Therefore, we invite you to submit a revised version of the manuscript that addresses the points raised during the review process. Please submit your revised manuscript within 60 days Apr 27 2025 11:59PM. If you will need more time than this to complete your revisions, please reply to this message or contact the journal office at digitalhealth@plos.org. Please include the following items when submitting your revised manuscript:* A rebuttal letter that responds to each point raised by the editor and reviewer(s). You should upload this letter as a separate file labeled 'Response to Reviewers '. This file does not need to include responses to any formatting updates and technical items listed in the 'Journal Requirements' section below.* A marked-up copy of your manuscript that highlights changes made to the original version. You should upload this as a separate file labeled 'Revised Manuscript with Track Changes '.* An unmarked version of your revised paper without tracked changes. You should upload this as a separate file labeled 'Manuscript '. If you would like to make changes to your financial disclosure, competing interests statement, or data availability statement, please make these updates within the submission form at the time of resubmission. Guidelines for resubmitting your figure files are available below the reviewer comments at the end of this letter. We look forward to receiving your revised manuscript. Kind regards, Cleva Villanueva, M.D., Ph.D.Academic EditorPLOS Digital Health Cleva VillanuevaAcademic EditorPLOS Digital Health Leo Anthony CeliEditor-in-ChiefPLOS Digital Healthorcid.org/0000-0001-6712-6626 **Journal Requirements:**

1. Please provide an Author Summary. This should appear in your manuscript between the Abstract (if applicable) and the Introduction, and should be 150–200 words long. The aim should be to make your findings accessible to a wide audience that includes both scientists and non-scientists. Sample summaries can be found on our website under Submission Guidelines:

https://journals.plos.org/digitalhealth/s/submission-guidelines#loc-parts-of-a-submission

 **Additional Editor Comments (if provided):** The manuscript has the potential to contribute significantly to the prevention of infectious diseases among homosexual men. However, it cannot be published in its current form in PLOS Digital Health.

Reviewers have raised several important concerns, comments, and suggestions, particularly regarding methodological issues and data requirements. It is essential to thoroughly address all the feedback provided by the reviewers to enhance the manuscript's quality**Reviewers' Comments:** Reviewer's Responses to Questions

**Comments to the Author**

1. Does this manuscript meet PLOS Digital Health’s publication criteria ? Is the manuscript technically sound, and do the data support the conclusions? The manuscript must describe methodologically and ethically rigorous research with conclusions that are appropriately drawn based on the data presented.

Reviewer #1: Yes

Reviewer #2: Yes

2. Has the statistical analysis been performed appropriately and rigorously?

Reviewer #1: Yes

Reviewer #2: Yes

3. Have the authors made all data underlying the findings in their manuscript fully available (please refer to the Data Availability Statement at the start of the manuscript PDF file)?

Reviewer #1: Yes

Reviewer #2: No

4. Is the manuscript presented in an intelligible fashion and written in standard English?

Reviewer #1: Yes

Reviewer #2: Yes

5. Review Comments to the Author

Reviewer #1: 1.Personally, I feel that the application of artificial functions in the prevention and control of infectious diseases may be a trend. Therefore, I think the selection of this research has certain application value.

2.The research design is feasible in general. However, the number of cases is small and the statistical method is simple; In this case, it is recommended to increase the sample size if possible.

3. The core concepts and methods in this article need to be explained or described in detail for the reader to understand.

Reviewer #2: See comments/suggestions:

Introduction

1. Add citations to this sentence: "Additionally, AI-driven chatbots have the potential to provide culturally competent and linguistically appropriate..."

2. Need to standardize how you refer to this: AI chatbots or AI-driven chatbots

Methods

3. Specify how many participants per focus group (range). Also, we all participants living in Southern US. Please clarify.

4. In p6, FGD has been spelled out in the study design and participants section but was not initially assigned the acronym.

5. Attach the interview guide as an appendix and make a citation where guide was mentioned.

6. Did you provide a brief overview of what AI-chatbots are as part of the FGD?

7. Specify tools mentioned in the brief survey. Like what depression screening tool was used.

8. I do understand that UTAUT was used as an analysis framework to scope the analysis. But, I want to check if there were themes that emerged from the data. I think it is important that qualitative studies do not limit themselves to a-priori themes so as to prevent limitations in interpreting participants' worldviews, especially if you want to compare data between US and Malaysia participants.

9. Provide more details on how you go about your qualitative analysis. Like data saturation and steps taken to achieve qualitative rigor (see details here: Shenton, A. K. (2004). Strategies for ensuring trustworthiness in qualitative research projects. Education for information, 22(2), 63-75.)

10. Qualitative scholars will feel bad to see thematic analysis as part of "statistical analyses." Statistical analyses is only applicable for quantitative data from the surveys but not the transcripts.

11. Easy to say that discrepancies were resolved but please provide proof of interrater reliability (Cohen's Kappa or krippendorff's Alpha) for at least one round of the review.

Results

12. Table 1 style is difficult to read since it is formatted into two columns. Much better to just do the traditional one-column table.

13. Can you at least let readers know the age and focus group number of the participants whose quote was used in the results.

14. To balance qualitative and quantitative perspectives, I find it helpful for authors to add numbers when using words that describe quantity, like most, some, few, etc. For instance "Most (70%; n = 15) believed that..."

15. This kind of sentences should be placed in the discussion since it is unclear if these were the results of analysis or a discussion of the findings: "Historical injustices and contemporary inequities in healthcare contribute to this mistrust..."

16. The placement of Figure 1 is so far from the point where it was mentioned.

Discussion

17. What does "Combined with other studies" mean? It does not really inform us how your results is related with previous work. I think it is better to use "Consistent with other studies," (or "In contrast with other studies" if your findings are opposite from previous work).

18. Create a subsection heading for study strengths and limitations since the last paragraph of your discussion reflects this.

Conclusion

19. Can you tone down this claim: "The use of AI chatbots will be an acceptable means of communication for behavioral change in relation to HIV prevention in general and PrEP uptake in particular." Perhaps limit this within the context of your study.

Others

20. Add details in funding that the funder had no role the specific aspects of the study.

6. PLOS authors have the option to publish the peer review history of their article (what does this mean? ). If published, this will include your full peer review and any attached files.

**Do you want your identity to be public for this peer review?** For information about this choice, including consent withdrawal, please see our Privacy Policy .

Reviewer #1: No

Reviewer #2: No

---

## [Decision Letter · Decision Letter 1]

13 May 2025

Formative Evaluation of the Acceptance of HIV Prevention Artificial Intelligence Chatbots by Black Gay, Bisexual, and Other Men Who Have Sex with Men in the Southern United States: Focus Group Study

PDIG-D-24-00512R1

Dear Jackson Jr Nforbewing Ndenkeh,

We are pleased to inform you that your manuscript 'Formative Evaluation of the Acceptance of HIV Prevention Artificial Intelligence Chatbots by Black Gay, Bisexual, and Other Men Who Have Sex with Men in the Southern United States: Focus Group Study' has been provisionally accepted for publication in PLOS Digital Health.

Best regards,

Cleva Villanueva, M.D., Ph.D.

Academic Editor

PLOS Digital Health

**Additional Editor Comments (if provided):**

The authors adequately addressed all reviewer comments, one of the reviewers approved the revised version, and the manuscript fulfills all the requirements for publication in PLOS Digital Health

**Reviewer Comments (if any, and for reference):**

Reviewer's Responses to Questions

**Comments to the Author**

1. If the authors have adequately addressed your comments raised in a previous round of review and you feel that this manuscript is now acceptable for publication, you may indicate that here to bypass the “Comments to the Author” section, enter your conflict of interest statement in the “Confidential to Editor” section, and submit your "Accept" recommendation.

Reviewer #2: All comments have been addressed

2. Does this manuscript meet PLOS Digital Health’s publication criteria ? Is the manuscript technically sound, and do the data support the conclusions? The manuscript must describe methodologically and ethically rigorous research with conclusions that are appropriately drawn based on the data presented.

Reviewer #2: Yes

3. Has the statistical analysis been performed appropriately and rigorously?

Reviewer #2: N/A

4. Have the authors made all data underlying the findings in their manuscript fully available (please refer to the Data Availability Statement at the start of the manuscript PDF file)?

Reviewer #2: Yes

5. Is the manuscript presented in an intelligible fashion and written in standard English?

Reviewer #2: Yes

6. Review Comments to the Author

Reviewer #2: Thank you for addressing the comments/suggestions.

7. PLOS authors have the option to publish the peer review history of their article (what does this mean? ). If published, this will include your full peer review and any attached files.

**Do you want your identity to be public for this peer review?** For information about this choice, including consent withdrawal, please see our Privacy Policy .

Reviewer #2: No
